# Cytotoxic CD8^+^ T Cells Are Involved in the Thrombo-Inflammatory Response during First-Diagnosed Atrial Fibrillation

**DOI:** 10.3390/cells12010141

**Published:** 2022-12-29

**Authors:** Julian Friebel, Marco Witkowski, Max Wegner, Leon Blöbaum, Stella Lammel, Philipp-Alexander Schencke, Kai Jakobs, Marianna Puccini, Daniela Reißner, Daniel Steffens, Verena Moos, Heinz-Peter Schutheiss, Ulf Landmesser, Ursula Rauch

**Affiliations:** 1Charité Center 11—Department of Cardiology, Charité—Universitätsmedizin Berlin, Corporate Member of Freie Universität Berlin and Humboldt-Universität zu Berlin, 12203 Berlin, Germany; 2Berlin Institute of Health at Charité—Universitätsmedizin Berlin, Charitéplatz 1, 10117 Berlin, Germany; 3DZHK (German Centre for Cardiovascular Research), Partner Site Berlin, 10785 Berlin, Germany; 4Department of Cardiac Anesthesiology and Intensive Care Medicine, German Heart Center, 13353 Berlin, Germany; 5Department of Cardiovascular & Metabolic Sciences, Lerner Research Institute, Cleveland Clinic, Cleveland, OH 44195, USA; 6Medical Department I, Gastroenterology, Infectious Diseases and Rheumatology, Charité—Universitätsmedizin Berlin, Corporate Member of Freie Universität Berlin and Humboldt-Universität zu Berlin, 12203 Berlin, Germany; 7Institute for Cardiac Diagnostics and Therapy (IKDT), 12203 Berlin, Germany

**Keywords:** atrial fibrillation, atrial myopathy, thrombo-inflammation, thrombin, protease-activated receptor, PAR1, tissue factor, t cells, CD8, MACE

## Abstract

Background: Atrial myopathy and atrial fibrillation (AF) accompany thrombo-inflammation. This facilitates disease progression and promotes major adverse cardiovascular events (MACEs). Thrombin receptor (protease-activated receptor 1, PAR1) signalling is central in mediating thrombo-inflammation. We hypothesised that PAR1 signalling links coagulation and inflammation through cytotoxic CD8^+^ T lymphocytes in patients presenting with first-diagnosed AF (FDAF). Methods: A total of 210 patients were studied. We included data and blood samples from patients presenting with FDAF (*n* = 160), cardiac tissue from patients with paroxysmal AF (*n* = 32) and 20 controls. Results: During early AF, a pro-inflammatory and cytotoxic subset of T lymphocytes (CD8^+^) circulated more frequently when compared to patients with chronic cardiovascular disease but without AF, accompanied by elevated plasma levels of CD8^+^ effector molecules, which corresponded to biomarkers of adverse cardiac remodelling and atrial dysfunction. Activation of tissue factor (TF) and PAR1 was associated with pro-inflammatory and cytotoxic effector functions. PAR1-related CD8^+^ cell activation was more frequent in FDAF patients that experienced a MACE. Conclusions: In patients with FDAF, the TF-factor Xa-factor IIa-axis contributes to thrombo-inflammation via PAR1 in CD8^+^ T cells. Intervening in this cascade might be a promising synergistic approach to reducing disease progression and the vascular complications of AF.

## 1. Introduction

Atrial fibrillation (AF) is the most common sustained cardiac arrhythmia in adults worldwide and is associated with substantial morbidity and mortality. Since age is a prominent AF risk factor, extended longevity will raise AF prevalence, which is currently between 2 and 4% in adults [1,2]. The first diagnosis of AF (FDAF) unveils a vulnerable, high-risk group of patients that needs continuous clinical monitoring and potential strategies beyond early rhythm control [3].

Other than age and genetic factors, modifiable risk factors—including hypertension, diabetes, heart failure, coronary artery disease, chronic kidney disease and obesity—contribute to the development and progression of adverse left atrial (LA) remodelling. LA dilation and fibrosis delay electromechanical conduction and are, thus, the substrate for AF. The burden of structural and electrophysiological LA remodelling correlates with the onset and intensity of AF. The complex atrial alterations associated with both atrial cardiomyopathy (or atrial myopathy, AM) and AF (endothelial activation, inflammation, oxidative stress, fibrosis, flow changes due to hyper-contractility) determine the thrombotic risk, which is responsible for cardioembolic stroke and peripheral ischaemic events [1,2,4,5].

Driven by cardiovascular risk factors, the concept of thrombo-inflammation describes this complex interplay between atrial fibrosis, inflammation and hyper-coagulability/thrombo-genicity [6,7,8]. This has been reported to play a critical role in the development and progression of AM, AF and major adverse cardiovascular events (MACEs) [4,9].

Activation of protease-activated receptor 1 (PAR1) signalling is central in mediating thrombo-inflammation [6,7,8,10,11,12]. Protease-activated receptors (PARs) comprise a family of four G protein-coupled receptors (PAR1–PAR4) that are activated through proteolytic cleavage by proteases derived from the coagulation cascade, including factor (F) Xa and thrombin (FIIa), as well as by immune cells and pathogens. Endothelial and subendothelial tissue factors (TFs) are upstream of FXa. PAR1 is the thrombin receptor on platelets, thereby coupling plasmatic with primary haemostasis. PAR1 is also expressed in endothelial cells, vascular smooth muscle cells, cardiac fibroblasts, cardiomyocytes and immune cells (monocytes and T cells) [13,14,15,16,17]. Thus, beyond its effect on coagulation, signalling via PAR1 has been linked to the pathogenesis of atherosclerosis, cardiovascular inflammation, cardiac fibrosis, myocarditis and heart failure [6,7,8,18,19,20,21,22,23,24,25,26].

Cytotoxic CD8^+^ T cells mediate myocardial damage in patients with virus-induced myocarditis and plaque erosion during acute coronary syndrome, whilst also promoting adverse post-ischaemic cardiac remodelling [23,27,28]. Upon activation, CD8^+^ T cells induce apoptosis of the target cell through the release of cytotoxins (perforin, granzymes and granulysin) or via cell-surface interaction (Fas ligand, FasL/receptor interactions). Their effector function is also executed by the secretion of pro-inflammatory cytokines, such as tumour necrosis factor α (TNF-α) and interferon-γ (IFN-γ) [29,30]. FIIa enhances the effector function of cytotoxic T cells through the activation of PAR1 [17,31,32]. There is emerging evidence supporting a link between T cell-mediated inflammation and AF disease progression [33,34,35,36]. The role of CD8^+^ T cells and PAR1 in AM—and especially early AF—has not been studied.

Therefore, we aimed to provide evidence of PAR1 signalling linking coagulation and inflammation mediated by CD8^+^ T lymphocytes in patients presenting with early AF.

## 2. Materials and Methods

### 2.1. Patient Studies

The local ethics committee (Charité—Universitätsmedizin Berlin) approved the study protocols, which were performed in accordance with the ethical principles in the Declaration of Helsinki. Each patient provided written, informed consent before participating in the study.

Routine laboratory results that are not mentioned separately in the methods section were obtained from medical records.

#### 2.1.1. Patients with First-Diagnosed AF and Control Group

Our main cohort consisted of 100 consecutive patients. Inclusion criteria for the AF group (*n* = 80) were as follows: age ≥ 18 years; willingness to sign a written informed consent form; admission to our cardiology department due to first-documented diagnosis of AF. Exclusion criteria for the AF group were as follows: reversible cause of AF (e.g., hyperthyroidism, acute myocardial infarction, myocarditis, pericarditis, acute infectious disease and acute inflammatory disease) or previous anti-coagulation. Baseline peripheral blood and data were collected directly within 24 h of hospitalisation.

During follow-up, MACE was defined as the occurrence of cardiovascular death, unplanned re-hospitalisation for AF, un-planned hospitalisation for heart failure, transient ischaemic attack (TIA), ischaemic stroke, acute coronary syndrome (ACS), deep vein thrombosis (DVT) or peripheral thromboembolism.

The control group (*n* = 20) consisted of consecutive patients with a comparable cardiovascular risk profile (but without AF) who had been admitted to our cardiology department. These patients were hospitalised due to hypertensive heart disease, heart failure or elective coronary angiography. During follow-up (one year), no patient in the control group developed AF or MACE.

A detailed description of each patient’s characteristics is provided in Table 1. Furthermore, we considered gender-specific aspects (Appendix A).

#### 2.1.2. Patients with Paroxysmal AF (Endomyocardial Biopsy)

Thrombo-inflammation in cardiac tissue was studied in a second cohort. Data from patients with paroxysmal AF (*n* = 32) were retrospectively generated from a data bank of the collaborative research network SFB19. Data were collected from un-selected patients who were available for sampling, with no adjustment for confounding factors. Patients underwent transvascular right ventricular endomyocardial biopsy (EMB) for histological, immuno-histological and virological examination because of suspected cardiomyopathy. For this, total cardiac mRNA was isolated from each biopsy, and the expression of PAR1 and TF was determined via quantitative PCR (qPCR). For qPCR, total mRNA was isolated using peqGOLD Trifast. The expression of the indicated markers was analysed with a FAM-tagged TaqMan^®^ gene expression assay. Relative gene expression was determined using the comparative C(t) (∆∆Ct) method with 18S ribosomal RNA as the endogenous control. Formalin-fixed and paraffin-embedded EMBs were analysed immuno-histologically to determine the distribution of perforin-expressing cells and fibrotic area. The patients were included if histological analysis of the biopsy sample showed no evidence of infiltrative or inflammatory myocardial disease. Patients were diagnosed with heart failure with preserved ejection fraction (HFpEF), hypertensive heart disease, ischemic cardiomyopathy or tachycardia-induced cardiomyopathy.

#### 2.1.3. Patients on Anti-Thrombotic Therapy after First Diagnosis of AF

Our third cohort consisted of separate patients (*n* = 80) (independent of first cohort) who have been on stable anti-thrombotic therapy after their first AF diagnosis (mean age 71.3 years, mean CHA_2_DS_2_-VASc-Score 3.85, mean HAS-BLED-Score 1.2, mean BMI 28.35 kg/m², female/male 41/39). Twenty patients received non-PAR-directed therapy (acetylsalicylic acid, ASA/vitamin K antagonist, VKA). Sixty patients received PAR-affecting therapy (FXa/FIIa-inhibitor, rivaroxaban/dabigatran). Peripheral blood and data were collected during an elective follow-up visit at our outpatient clinic. MACE was defined as the occurrence of cardiovascular death, un-planned re-hospitalisation for AF, un-planned hospitalisation for heart failure, TIA, ischaemic stroke, ACS, DVT and peripheral thromboembolism.

### 2.2. ELISA

Granulysin, granzyme A, granzyme B, soluble FasL (sFasL), soluble interleukin 1 receptor-like 1 (sST2), galectin 3, N-terminal pro-atrial natriuretic peptide (NT-proANP) and TF ELISA (R&D Systems, Minneapolis, MN, USA) were performed according to the manufacturer’s instructions.

### 2.3. Flow Cytometry

Peripheral blood mononuclear cells (PBMCs) were prepared as previously described [37]. Thawed PBMC were fixed for 10 minutes with 4% formalin followed by staining with anti-human CD3 PE (Becton, Dickinson and Company, Franklin Lakes, NJ, USA) and PAR1-Alexa488 (R&D Systems, Minneapolis, MN, USA). The percentage of FIIa-activated PAR1-positive (the antibody detects fragments of activated PAR1-cleaved-Ser^42^ protein) (Sigma-Aldrich, St. Louis, MO, USA) circulating CD8^+^ T cells (anti-human CD8 PerCP from Becton, Dickinson and Company, Franklin Lakes, NJ, USA) was measured using flow cytometry with a FACSCalibur^TM^ and CellQuest software (both Becton, Dickinson and Company, Franklin Lakes, NJ, USA) [18,19,38]. The intracellular staining of cytokines after PBMC stimulation was performed as previously described [37,39].

### 2.4. Statistical Analysis

Single comparisons were assessed using the Mann–Whitney test. For correlation analysis, the Pearson coefficient was used. All analyses were performed using GraphPad Prism version 9.3.0 software. Results are expressed as single values ± SD. The overall α-level was 0.05.

## 3. Results

### 3.1. Activation of CD8^+^ T Cells in Patients with First-Diagnosed AF

The activation of T lymphocytes has been observed in patients with more advanced AF phenotypes (e.g., in paroxysmal, long-standing, persistent and permanent AF) [33,34]. However, their role in patients with FDAF—subjects in whom secondary preventive efforts to modify the course of the disease are critical—remains poorly understood. Here, we report that in this early stage of AF, a pro-inflammatory and cytotoxic subset of T cells (CD8^+^) circulates more frequently when compared to control patients with chronic cardiovascular disease but without AF (Figure 1A). These T cells express the surface activation marker HLD-DR (Figure 1A). The surface marker CD28 characterises naive/early-differentiated T cells, which arise during acute immune responses. In contrast, CD57 is expressed by mature T cells, indicating chronic immune activation [29,30]. In our study, the percentage of circulating mature CD8^+^CD57^+^ cells, but not of naive/early CD8^+^CD28^+^ cells, increased when compared to the control group (Figure 1A). Likewise, plasma levels of cytotoxic effector molecules (granulysin, granzymes and sFasL) were more abundant during FDAF when compared to the control group (Figure 1B).

### 3.2. Activation of CD8^+^ T Cells Correlates with Biomarkers of Cardiac Fibrosis and Atrial Dysfunction in Patients with First-Diagnosed AF

AM is characterised by adverse atrial remodelling due to increased collagen deposition [2,4,5]. The biomarkers sT2, galectin-3 and NT-proANP indicate cardiac fibrosis and atrial dysfunction [40,41]. In our patient cohort with a first diagnosis of AF, plasma levels of sT2, galectin-3 and NT-proANP were higher than in the control subjects (Figure 2A). Furthermore, cytotoxic effector molecules (granulysin, granzymes and sFasL) positively correlated with the plasma levels of these surrogate markers of AM in our study (Figure 2B). This suggests a potential link between CD8^+^ T cell activation, cardiac fibrosis and AM.

### 3.3. CD8^+^ T Cell-Mediated Effector Function in Early AF Is Linked to PAR1 Activation

Endothelial and subendothelial TF initiates the coagulation cascade that maintains a pre- and pro-thrombotic state and promotes pro-inflammatory PAR signalling [25,42]. This has already been shown in the early stages of arrhythmia [43,44,45,46]. The next immediate question for consideration is the following: Is the TF pathway linked to the observed pro-inflammatory and cytotoxic activity in patients with FDAF? In our study, we demonstrated that elevated levels of circulating TF associated with biomarkers are indicative of an increase in CD8^+^ T cell activation (Figure 3).

Thus far, we have demonstrated that cytotoxic T cell effector molecules are associated with circulating biomarkers of cardiac fibrosis and atrial dysfunction in patients with FDAF. To address whether PAR1 is a putative mediator of TF-initiated cardiac T cell activation and fibrosis, we analysed the EMBs of patients with AF. The transcription levels of TF and PAR1 were positively associated (Figure 4A). We found that the transcription of PAR1 correlated with the distribution of perforin-expressing cells infiltrating the cardiac tissue (Figure 4B). Furthermore, an increase in cardiac PAR1 expression correlated with higher cardiac collagen deposition (Figure 4B). This suggests that cardiac PAR1 signalling is associated with cytotoxic T cell activation and adverse structural remodelling in patients with AF.

Signalling via PAR1 is a presumed link between the coagulation system (TF-FXa/FIIa-PAR-axis) and the T cell effector function (=thrombo-inflammation) [31,32]. FIIa-activated PAR1-positive CD8^+^ T cells (the antibody detects activated PAR1 cleaved by FIIa) were used as an indicator for augmented signalling through the TF-FXa/FIIa-PAR1 axis [18,19,38]. In patients with FDAF, T cells that express PAR1 were more frequent when compared to control subjects (Figure 5A). In FDAF patients, the distribution of thrombin-activated PAR1 in T cells was enhanced (Figure 5A). Further phenotyping revealed that CD8^+^ T cells specifically showed signs of thrombin activation, since the percentage of cytotoxic T cells expressing cleaved PAR1 (=thrombin-activated PAR1) was higher in patients with early AF when compared to patients without AF (Figure 5A).

Next, we tested the functional capacity of CD8^+^PAR1^+^ T cells from patients presenting with their first AF episode. An increase in TNF-α and IFN-γ production after polyclonal stimulation reveals the pro-inflammatory effector function of this T cell subpopulation (Figure 5B).

Previous studies have demonstrated that PAR-directed therapeutic intervention reduces T cell activation [18,31,32]. Therefore, we tested the hypothesis that a PAR-affecting therapy reduces the pro-inflammatory response of CD8^+^PAR1^+^ T cells. Our third cohort consisted of separate patients (independent of the first cohort) on stable anti-thrombotic therapy after FDAF. The frequency of pro-inflammatory CD8^+^PAR^+^ T cells (as expressed by positivity for TNF-α and IFN-γ) was higher in AF patients on a non-PAR-directed therapy (ASA/VKA) when compared to the group receiving a PAR-affecting therapy (FXa/FIIa-inhibitor) (Figure 5C).

### 3.4. Frequency of Thrombin-Activated CD8^+^PAR1^+^ T Cells Is Associated with Adverse Outcomes in Patients with First Diagnosis of AF

Thus far, we have described a pro-inflammatory and cytotoxic milieu during early AF. Notably, AF is associated with cardiovascular morbidity and mortality [1,3]. In our cohort of patients with first-diagnosed AF, 42.5% experienced MACEs during a mean follow-up of 2.99 years. Patients who suffered from an adverse event after their first AF diagnosis showed a higher frequency of thrombin-activated CD8^+^PAR1^+^ T cells at baseline when compared to patients who did not develop MACEs during follow-up (Figure 6A). The disease progression of AM/AF (as expressed by un-planned re-hospitalisation for AF and un-planned hospitalisation for heart failure) and thrombo-embolic and atherothrombotic events (as expressed by the onset of TIA/ischaemic stroke and ACS) corresponded to the frequency of circulating CD8^+^PAR1^+^ T cells at baseline (Figure 6B,C).

We tested our hypothesis that the activation of PAR1 in cytotoxic T cells is associated with the incidence of MACE in our third cohort of 80 AF patients (independent of the first cohort) that were already on stable anti-thrombotic therapy after their first AF diagnosis. In this cohort (after a mean follow-up of 1.77 years), we observed a higher percentage of activated CD8^+^PAR^+^ T cells at follow-up in the group that had suffered from MACEs when compared to FDAF patients without MACEs (% CD8^+^ cleaved PAR1^+^ cells MACE, no vs. yes: 50.3 ± 17.7 vs. 63.6 ± 11, *p* = 0.0008).

These data suggest that thrombo-inflammation mediated by CD8^+^ T cells is associated with the onset of adverse outcome events in patients after FDAF.

## 4. Discussion

Inflammation and atrial fibrosis are fundamental mechanisms of the pathophysiological processes involved in the development of AM and AF. The degree of atrial inflammation and fibrosis correlates with the susceptibility to AF, increases the frequency of AF paroxysms and increases the likelihood of progression to persistent and permanent AF [1,47].

By studying three cohorts of patients with AF, we discovered the following:The activation of CD8^+^ T cells is present in patients with FDAF.Cardiac fibrosis and atrial dysfunction correlate with CD8^+^ T cell activation.CD8^+^ T cell-mediated effector function in early AF is linked to PAR1 activation.The frequency of thrombin-activated CD8^+^PAR1^+^ T cells is associated with adverse outcomes in patients with FDAF.PAR-directed therapeutics (FIIa/FXa-inhibitors) mediate pleiotropic effects by targeting T cell effector function.

### 4.1. Activation of CD8^+^ T Cells in Patients with First-Diagnosed AF

Previous studies have highlighted the important role of cellular immunity in the pathogenesis of AM and AF [35,36,48]. These studies were conducted in patients with paroxysmal, persistent, long-standing persistent and permanent AF but not explicitly in patients with FDAF, for whom understanding the disease pathogenesis is critical to guiding early secondary preventive efforts. These previous studies reported a higher degree of circulating CD3^+^ T lymphocytes expressing the surface activation marker HLD-DR [33,34]. The activation of T cells, as expressed by HLA-DR positivity, remained high in patients with the recurrence of AF after a successful cardioversion when compared to patients that continued in sinus rhythm during a 3-month follow-up [34]. A reduced number of regulatory T cells suggests that this activation is not sufficiently counterbalanced in AF patients [49,50,51].

CD8^+^ T cells undergo distinct phases of maturation. Naive CD8^+^ T cells that express the surface protein CD28 arise from common lymphoid progenitors that migrate from the bone marrow to the thymus. Their initial priming requires interaction with professional antigen-presenting cells. Further priming, proliferation and activation depend on cytokines that are secreted from CD4^+^ T cells [29,30]. T cell phenotyping revealed a higher number of this subset in patients with AF [48,49]. Repeated rounds of T cell activation due to antigen stimulation occur during long-term chronic inflammation. This leads to the downregulation of CD28 and the upregulation of CD57 surface proteins. Therefore, CD8^+^CD28^−^CD57^+^ T lymphocytes represent a late or terminally differentiated T cell line (matured). This subset possesses a pro-inflammatory potential characterised by the production of IFN-γ and TNF-α as well as a cytotoxic potential mediated by perforin, granulysin and granzymes. Oligoclonal expansion and resistance to apoptosis led to an accumulation of this T cell subset over time [29,30]. The increase in the CD8^+^CD28^−^CD57^+^ subset, in contrast to the CD28^+^, suggests a chronic rather than an acute or reactive immune stimulus. Circulating CD8^+^ T cells exhibit no elevated expression of programmed cell death protein-1 (PD-1) in patients with AF lasting < 1 year [33]. PD-1 expression indicates T cell exhaustion and susceptibility to apoptosis, which are associated with decreased effector function [29,30]. However, two studies conducted on patients that underwent elective cardiac surgery support the notion that T cell-mediated inflammatory and cytotoxic effector function contribute to the new onset of AF in the perioperative setting [52,53].

The next immediate question that arises is whether T cell-mediated inflammation is only a systemic phenomenon or, simultaneously, an active inflammatory process in the heart during AF [54]. Previous biopsy studies have highlighted that the number of CD3^+^ T cells infiltrating the LA appendage tissue, LA myocardium and surrounding adipose tissue was already higher in patients with paroxysmal AF when compared to patients with sinus rhythm [54,55,56,57]. Notably, the degree of LA dilation—an indicator of atrial cardiomyopathy that is also associated with atrial fibrosis—correlated with the extent of infiltrating CD3^+^ T cells [56,57]. Another study found that fibrotic areas of subepicardial fatty infiltrates (of the LA) pre-dominantly contained CD8^+^ T cells. Their cytotoxic activity was indicated by positive staining with granzyme [58].

Our results suggest an association of cardiac PAR1 expression with markers of thrombo-inflammation in patients with AF. However, it must be mentioned that the results obtained from cardiac tissue samples might be influenced by the underlying pathology, as well as by the method and location of sampling.

### 4.2. CD8^+^ T Cell-Mediated Effector Function in Early AF Is Linked to PAR1 Activation

The transition from sub-clinical to clinical AF (which leads to the first diagnosis of AF) is often characterised by advancing atrial structural remodelling or the worsening of AM [1,2,4,59]. Following endothelial activation (which is already present during early AF) with the upregulation of adhesion molecules (and due to chemotactic signals), CD8^+^ T cells traffic to peripheral tissues and exert effector functions, including cytotoxicity and cytokine secretion [45,60]. Infiltrating CD8^+^ immune cells can indirectly influence electrical conduction in the LA via crosstalk with fibroblasts and cardiomyocytes. The release of inflammatory cytokines (e.g., TNF-α and IFN-γ) initiates pro-fibrotic pathways in fibroblasts and mediates cardiomyocyte dysfunction [61]. Cytotoxic granules foster cardiomyocyte death through the local release of perforin, granulysin and granzymes, which is reflected by higher values of troponin in patients with AM and AF [27,61,62,63,64,65,66,67]. The pro-inflammatory and cytotoxic activity of CD8^+^ T cells has previously been linked to all stages of atherosclerosis, the pathogenesis of ACS (especially due to plaque erosion), post-ischaemic adverse remodelling and acute myocarditis [23,27,28,68,69].

Other than maintaining haemostasis and vascular integrity, the coagulation proteases FXa and FIIa mediate endothelial activation and vascular inflammation (=thrombo-inflammation). Notably, they also affect adverse cardiac remodelling. The cellular mechanisms initiated by the coagulation proteases are mediated by PARs, which are a ubiquitously expressed class of G-protein-coupled receptors. Therefore, PARs are critical regulators of platelet function, drive inflammatory responses, mediate fibrosis and lead to the dysfunction of cardiomyocytes [6,7,8,10,11,12,18,19,21,22,23,26]. Both AM and the underlying cardiovascular risk factors synergistically drive endothelial activation. Moreover, AF itself aggravates this state [2,4,9]. Consequently, endothelial and subendothelial-derived TF initiates the coagulation cascade that maintains a pre- and pro-thrombotic state [42,70,71]. Notably, markers of hypercoagulability are elevated, even in the early stages of the arrhythmia [43,44,45,46].

Both CD4^+^ and CD8^+^ T cells express PAR1 (CD8^+^ > CD4^+^) [17]. PAR1 expression levels increase with CD8^+^ T cell maturation/differentiation. Two previous studies have shown that especially antigen-experienced (late/terminally differentiated = matured) CD8^+^ T cells that migrate to peripheral tissues have an enhanced expression of PAR1 when compared to naive subsets. Thrombin induced inflammatory IFN-γ cytokine secretion by CD8^+^ T cells. PAR1 expression also correlated with the expression of the effector molecules granzyme and perforin, thereby highlighting their increased cytotoxic potential [31,32].

### 4.3. Clinical Implications

AF is a chronic inflammatory disease in which atherothrombotic complications lead to cardiovascular morbidity and mortality. AF can induce MACEs, thereby resulting in a higher risk of stroke, worsening heart failure and ACS [1,3].

In this study, we demonstrated that the frequency of thrombin-activated CD8^+^PAR1^+^ cytotoxic T cells at the time of FDAF was associated with MACEs during follow-up. CD4^+^CD28^−^ T cells, which represent another subset with cytotoxic properties, predicted mortality in patients with AF and heart failure [49]. A high frequency of CD8^+^CD57^+^ T cells was associated with short-term cardiovascular mortality in acute myocardial infarction (MI) patients [69]. In line with this observation, acute MI patients with high circulating levels of granzyme B (when compared to patients with low levels) were at higher risk of death after 1 year of follow-up [27]. Notably, it has been shown that patients with first-diagnosed and early AF (compared to patients with non-paroxysmal stages) have an especially increased risk for ACS [3,72,73]. An early rhythm control strategy—which is particularly suggested for patients with a high comorbidity burden (e.g., history of heart failure, CHA_2_DS_2_-VASc score ≥ 4)—was not beneficial with regard to hospitalisation and ACS during follow-up in patients with first-diagnosed AF [3,74,75]. This highlights the fact that the first diagnosis of AF unveils a vulnerable, high-risk group of patients that needs continuous clinical monitoring and potential strategies beyond early rhythm control [3].

Endothelial dysfunction and vascular inflammation are essential in the development of acute cardiovascular events. We suggest that CD8^+^ T cell-related inflammation and cytotoxicity are not only restricted to LA but might also contribute to the pathogenesis of acute atherothrombotic events, as shown for the pathogenesis of ACS following plaque erosion [28]. Furthermore, CD8^+^ T cell activation is linked to the development and progression of heart failure, thereby potentially reducing myocardial ischaemic resistance during AF paroxysms [27]. Following an acute MI, anti-inflammatory and anti-thrombotic therapy have been suggested for secondary prevention in patients with residual inflammatory or residual thrombotic risk, respectively [76].

PARs are important regulators of various physiological responses and are implicated in numerous pathological conditions [13]. PAR-targeting drugs (non-vitamin K antagonist oral anticoagulants, NOACs) possess pleiotropic (non-haemostatic) effects that address the increased (PAR-mediated) thrombo-inflammatory burden in patients with cardiovascular diseases. They have been shown to reduce atherosclerosis, acute atherothrombotic events, cardiac fibrosis and cardiac dysfunction [6,7,12,19]. Furthermore, the direct PAR1-inhibitor vorapaxar reduced T cell-related inflammation markers in a pre-clinical model of atherosclerosis [18]. A previous study also showed that upstream PAR inhibition via targeting FXa (compared to VKA) reduced PAR expression in the circulating immune cells of patients with HFpEF [19]. Coagulation factors synthesised during VKA treatment retain their capability to activate PARs, whereas NOAC treatment inhibits both the coagulation and activation of PARs [11,19,38,77]. Here, we demonstrate that a PAR-directed therapy was associated with a reduced pro-inflammatory immune response of cytotoxic T cells. In this regard, FXIa-inhibition (beyond NOACs) might be an interesting new therapeutic option [78,79].

### 4.4. Limitations

Although the first diagnosis of AF can be precisely defined, the individual AF duration is obscure. Therefore, based on our data, we cannot conclude at what point in time CD8^+^ T cell activation is involved in the pathogenesis of AM and the onset of AF. Patients with AF were studied in a retrospective analysis with inherent biases and limitations. Therefore, prospective trials that account for the different phenotypes of AF (or AM as its precursor), as well as the different therapeutic strategies (early rhythm control vs. rate control) and include cardiac magnetic resonance imaging and EMBs would be necessary.

## 5. Conclusions

Pro-inflammatory and cytotoxic CD8^+^ T cell activation characterises patients with first-diagnosed AF. In this group of patients, the TF-FXa-FIIa axis contributes to thrombo-inflammation via PAR1 in CD8^+^ T cells. Cytotoxic activity is linked to cardiac fibrosis and atrial dysfunction as the basis of AM. PAR1-related CD8^+^ cell activation associates with adverse cardiovascular events in patients following newly diagnosed AF. Therefore, intervening in the thrombo-inflammatory cascade might be a promising synergistic approach (e.g., anticoagulation and a pleiotropic, anti-inflammatory effect on CD8^+^ T cells) to reduce the disease progression of AF.

## Figures and Tables

**Figure 1 cells-12-00141-f001:**
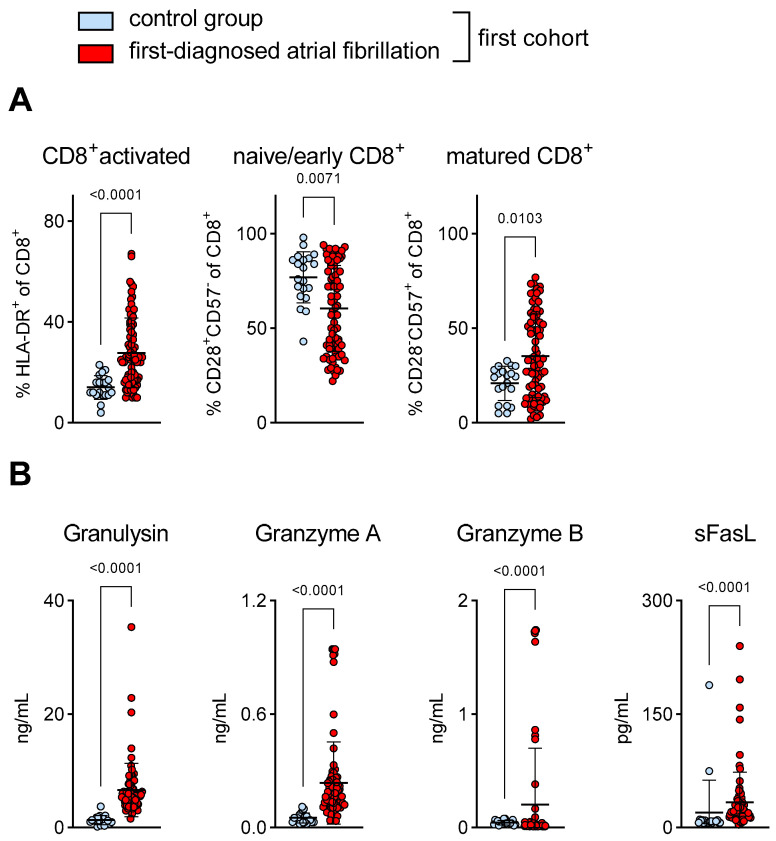
Activation of CD8^+^ T cells in patients with first-diagnosed AF. (**A**) Phenotyping of circulating CD8^+^ T cells (flow cytometry). Activated cytotoxic T cells (HLA-DR^+^), naive/early differentiated CD8^+^ T cells (CD28^+^CD57^−^) and matured CD8^+^ T cells (CD28^−^CD57^+^). (**B**) Increased plasma levels of cytotoxic effector molecules associated with CD8^+^ T cell function (ELISA). Results are expressed as single values (*n* = 20/80), mean with SD.

**Figure 2 cells-12-00141-f002:**
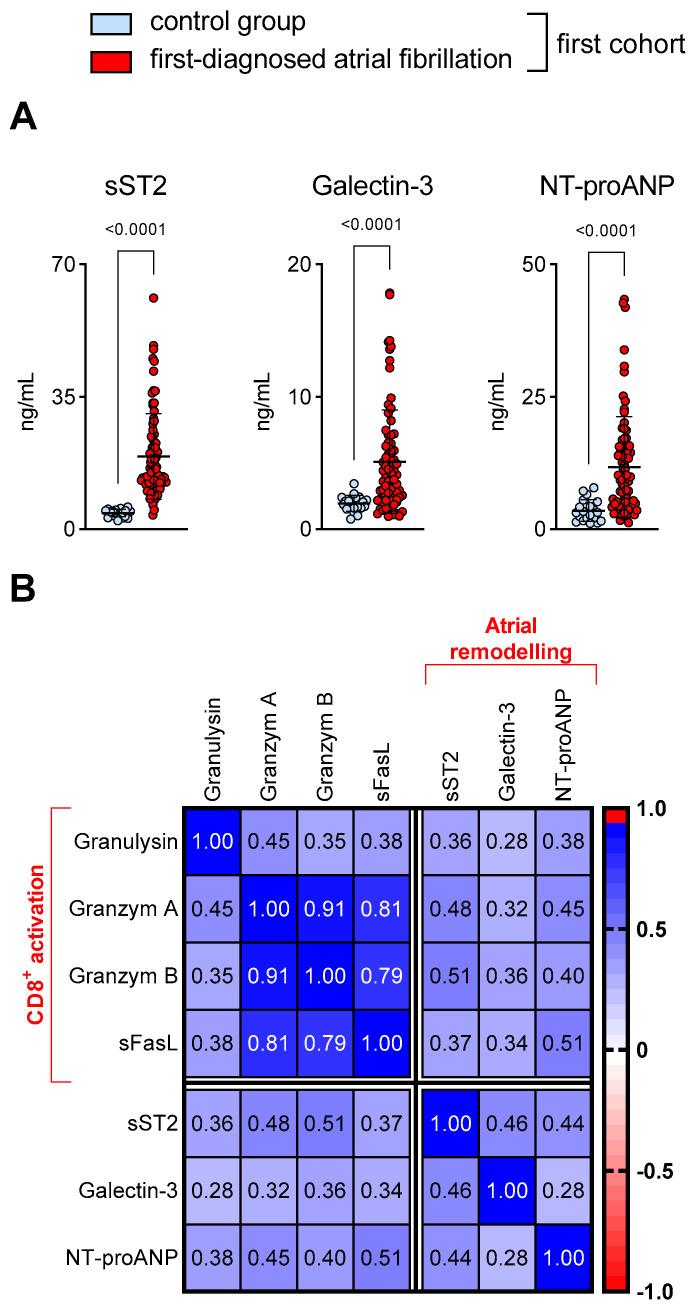
Activation of CD8^+^ T cells corresponds to biomarkers of cardiac fibrosis and atrial dysfunction in patients with first-diagnosed AF. (**A**) Increased plasma levels of biomarkers that suggest cardiac fibrosis (sST2, galectin-3) and atrial dysfunction (NT-proANP) (ELISA). Results are expressed as single values (*n* = 20/80), mean with SD. (**B**) Correlation matrix comparing CD8^+^ T cell effector molecules (granulysin, granzymes, sFasL) with sT2, galectin-3 and NT-proANP plasma levels in patients with first-diagnosed AF (*n* = 80). Pearson correlation coefficients (+1 positive, −1 negative correlation) are shown.

**Figure 3 cells-12-00141-f003:**
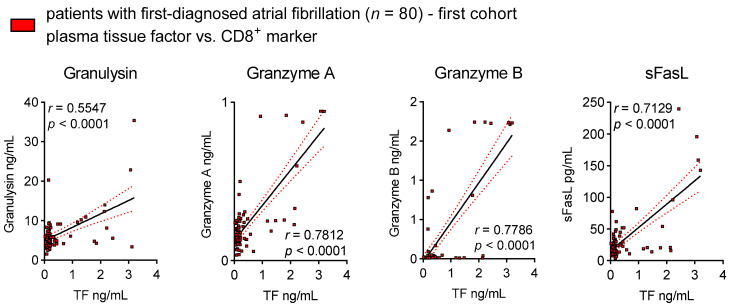
Thrombogenicity corresponds to CD8^+^ T cell activation. Plasma levels of tissue factor (TF) relative to plasma levels of cytotoxic effector molecules in patients with first-diagnosed AF (ELISA). Results are expressed as single values (*n* = 80), Pearson correlation coefficients and linear regression lines with 95% CI.

**Figure 4 cells-12-00141-f004:**
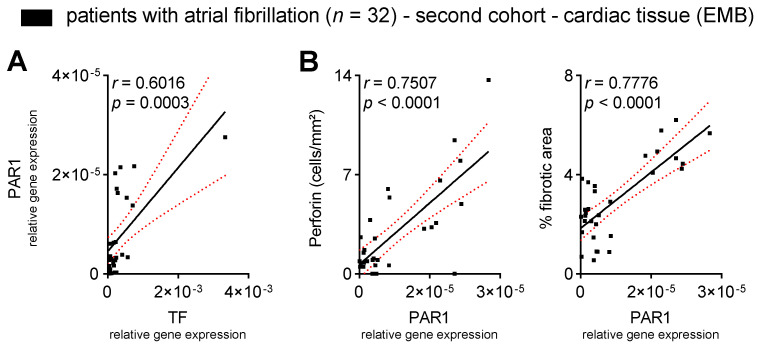
Cardiac PAR1 expression correlates with markers of thrombo-inflammation in patients with AF. Patients underwent transvascular right-ventricular endomyocardial biopsy (EMB). PAR1 gene expression in EMBs: (**A**) relative to TF gene expression; (**B**) relative to the distribution of perforin-positive cytotoxic T cells and the extent of cardiac fibrosis. Results are expressed as single values (*n* = 32), Pearson correlation coefficients and linear regression lines with 95% CI.

**Figure 5 cells-12-00141-f005:**
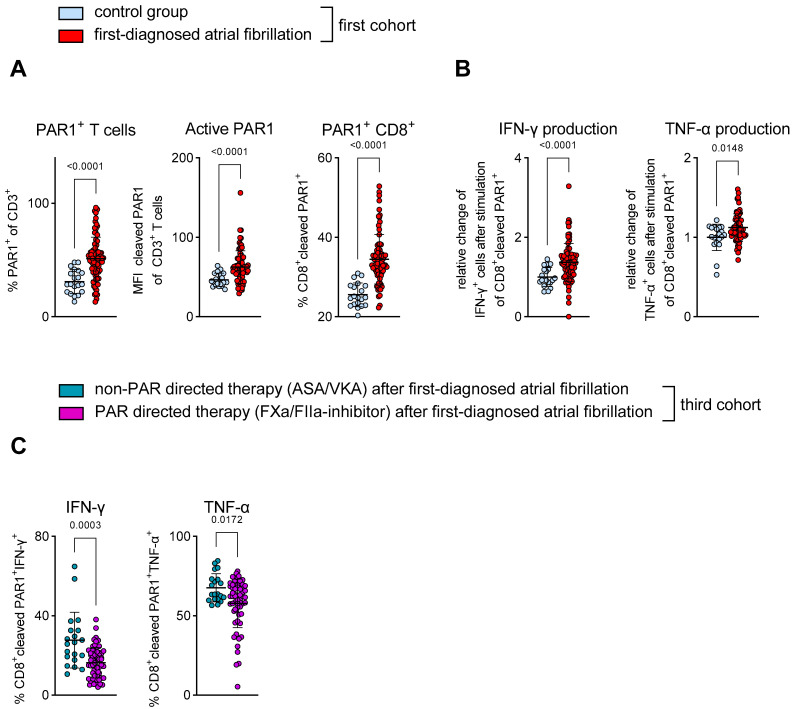
CD8^+^ T cell-mediated effector function in early AF is linked to PAR1 activation. (**A**,**B**) Patients with first-diagnosed AF vs. control group. (**A**) Phenotyping via flow cytometry of T cells in patients with first-diagnosed AF revealed a higher percentage of circulating cells that possess PAR1 (CD3^+^PAR1^+^), an increased expression of thrombin-activated PAR1 (as expressed by an increased mean fluorescence intensity (MFI) of thrombin-cleaved PAR1) and a higher percentage of cytotoxic T cells that express PAR1, which was activated by thrombin (CD8^+^ cleaved PAR1^+^). (**B**) Enhanced pro-inflammatory effector function of thrombin-activated CD8^+^PAR1^+^ T cells as expressed by staining for IFN-γ and TNF-α. (**C**) Separate cohort of patients (third cohort) on stable anti-thrombotic therapy. Lower percentage of circulating pro-inflammatory thrombin-activated CD8^+^PAR1^+^ T cells under a PAR-directed therapy (FXa/FIIa-inhibitor) when compared to non-PAR-directed therapy (ASA/VKA). Results are expressed as single values (A,B *n* = 20/80; C *n* = 20/60), mean with SD.

**Figure 6 cells-12-00141-f006:**
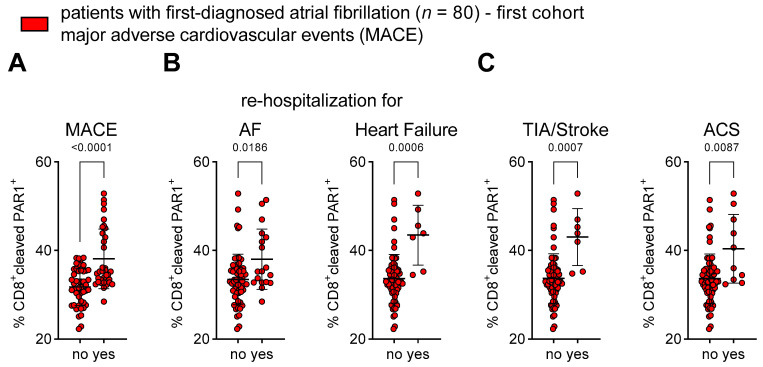
Frequency of thrombin-activated CD8^+^PAR1^+^ T cells is associated with adverse outcomes in patients with first diagnosis of AF. Patients were stratified into two groups according to the occurrence of MACEs (no vs. yes) during follow-up after their first diagnosis of AF. (**A**) Composite endpoint MACE: first occurrence of cardiovascular death, unplanned re-hospitalisation for AF, un-planned hospitalisation for heart failure (HF), transient ischaemic attack (TIA), ischaemic stroke, acute coronary syndrome (ACS), deep vein thrombosis, peripheral thromboembolism. (**B**,**C**) Adverse outcome events that are associated with the disease progression of AF (**B**) (un-planned re-hospitalisation for AF, un-planned hospitalisation for HF) and thrombo-embolic and atherothrombotic complications (**C**) (TIA, ischaemic stroke, ACS) are related to CD8^+^PAR1^+^ T cells at the time of first diagnosis of AF. Results are expressed as single values (*n* = 80), mean with SD.

**Table 1 cells-12-00141-t001:** Baseline characteristics of patients with first-diagnosed AF and controls. First cohort.

	Control Group	Patients with First-Diagnosed AF	
	(*n* = 20)	(*n* = 80)	*p*-Value
Male/Female	10/10 (50%/50%)	50/30 (62.5%/37.5%)	n.s.
CHA_2_DS_2_-VASc	3.45	3.98	n.s.
History of Heart Failure	4/20 (20%)	21/80 (26%)	n.s.
Ejection Fraction	62%	59%	n.s.
NT-pro BNP ng/L	170.3	2153	<0.0001
Hypertension	17/20 (85%)	70/80 (87.5%)	n.s.
Age (years)			
<65	9/29 (45%)	22/80 (27.5%)	n.s.
65–75	6/20 (30%)	26/80 (32.5%)	n.s.
>75	5/20 (25%)	32/80 (40%)	n.s.
Diabetes	5/20 (25%)	24/80 (30%)	n.s.
History of TIA/Stroke	1/20 (5%)	8/80 (10%)	n.s.
Body Weight (kg)	82.95	85.93	n.s.
BMI kg/m²	27.97	27.55	n.s.
PCT µg/L (ULN 0.09)	0.07	0.07	n.s.
CRP mg/L (ULN 5 mg/L)	3.33	3.25	n.s.

Values are expressed as the mean or *n* (%). Abbreviations: NT-pro BNP, N-terminal prohormone of brain natriuretic peptide; TIA, transient ischaemic attack; BMI, body mass index; ULN, upper limit of normal; PCT, procalcitonin; CRP, C-reactive protein; n.s., not significant.

## Data Availability

Data from patients are not publicly available due to general data protection regulations.

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
