# Peer review of "Cytotoxic CD8+ T Cells Are Involved in the Thrombo-Inflammatory Response during First-Diagnosed Atrial Fibrillation"

_cells, 2022, doi:10.3390/cells12010141_

Round 1

Reviewer 1 Report

The paper “Cytotoxic CD8+ T cells are involved in the thrombo-inflammatory response during first-diagnosed atrial fibrillation” is an elegant translational research paper whose working hypothesis was that cytotoxic CD8+ T lymphocytes link PAR-1 signaling to coagulation and inflammation in patients with first diagnosed AF. The Authors studied 80 consecutive patients presenting with first diagnosed AF, 32 unselected patients with paroxysmal AF who had undergone endomyocardial biopsy because of suspected cardiomyopathy whose data were retrospectively generated from a data bank of the collaborative research network SFB 19, and 80 patients with a history of AF (mean CHA2DS2-VASc score 3.85) who were on stable anti-thrombotic therapy and in whom MACEs were evaluated.

In the first cohort of patients, the Authors aimed to disclose whether in patients with first diagnosed AF the T-lymphocytes are activated as it has been observed in patients with more advanced AF phenotypes. They provide evidence that indeed in patients with newly diagnosed AF a pro-inflammatory and cytotoxic subset of T- lymphocytes (CD8+) circulates more frequently when compared with the control group of patients (n. 20) with a comparable cardiovascular risk profile and no AF. The Authors also found a positive correlation between cytotoxic effector molecules of CD8+ T cell activation and surrogate markers of atrial myopathy and myocardial fibrosis. Furthermore, the Authors demonstrated the association of elevated levels of circulating TF with the biomarkers of increased CD8+ T cell activation.

In the second cohort of patients, the Authors aimed to study thrombo-inflammation in right ventricular myocardial tissue by looking at the total cardiac mRNA and the expression of PAR1 and TF via quantitative PCR, whereas the distribution of perforin-expressing cells and the fibrotic area was analyzed by immunohistology on formalin-fixed and paraffin-embedded samples. The positive association of the transcription levels of TF and PAR1, and the correlation of the transcription of PAR1 with the distribution of perforin-expressing cells infiltrating the ventricular tissue leads the Authors to suggest that cardiac PAR1 signaling is associated with cytotoxic T cell activation and adverse structural remodeling in patients with AF.

MACEs, defined as the occurrence 134 of cardiovascular death, unplanned re-hospitalization for AF, unplanned hospitalization for heart failure, TIA, ischemic stroke, ACS, DVT, and peripheral thromboembolism, were evaluated in the 3rd cohort of patients. The Authors by analyzing blood samples of the 3rd cohort of patients who were on stable anti-thrombotic therapy after their first episode of AF. The Authors found that the frequency of pro- 2 inflammatory CD8+PAR+ T was higher in AF patients on a non-PAR-directed therapy (ASA/ VKA) when compared to the group receiving a PAR-affecting therapy (FXa/FIIa-inhibitor).

Which was the eventual diagnosis for the 32 patients of cohort n.2? My feeling is that the conclusions/clinical suggestions derived from the data of the second cohorts of patients are the weakest because biased by the main pathology which led to an endomyocardial biopsy, although “Patients were included if histological analysis of the biopsy sample showed no evidence of infiltrative or inflammatory myocardial disease”. Furthermore, ventricular tissue was analyzed, not atrial, so conclusions/suggestions are only inferential. This should be acknowledged in the discussion.

Is the 3rd cohort of patients the same as the first-diagnosed AF patients of cohorts n.1? This should be stated more clearly in the initial description of the population. More details about those 80 patients would be appreciated, particularly details on age and specific therapy.

Each set of results is introduced by the corresponding background (even quoting references) and working hypothesis to lead the reader to understand the process to final conclusion, which is rather unusual. Results are results.

Minor remark: the first cohort of patients with newly onset AF is 80 and not 100.

Author Response

Reviewer #1:

Comment 1:

Which was the eventual diagnosis for the 32 patients of cohort n.2? My feeling is that the conclusions/clinical suggestions derived from the data of the second cohorts of patients are the weakest because biased by the main pathology which led to an endomyocardial biopsy, although “Patients were included if histological analysis of the biopsy sample showed no evidence of infiltrative or inflammatory myocardial disease”.

Reply:

We thank the reviewer in supporting our intention and we understand the reviewer’s concern about the potential risk of bias. We had already stated in the methods section that data were collected from unselected patients who were available for sampling, with no adjustment of confounding.

We suggested that cardiac PAR1 signalling was associated with cytotoxic T cell activation and adverse structural remodelling in patients with AF. Our second cohort consisted of patients which were diagnosed with heart failure with preserved ejection fraction (HFpEF), hypertensive heart disease, ischemic cardiomyopathy, or tachycardia-induced cardiomyopathy. We have added this information to the methods section.  Of course, AF might only be seen as a bystander or confounder, but all conditions that were diagnosed in our EMB-cohort are known risk factors for the development of AF, or the consequence of AF itself, which, in the opinion of the authors, strengthens our results.  Nevertheless, we cannot exclude the risk of bias or confounding. Therefore, we now have discussed this limitation to address your concerns.

Comment 2:

Furthermore, ventricular tissue was analyzed, not atrial, so conclusions/suggestions are only inferential. This should be acknowledged in the discussion.

Reply:

Several statements that we made were more ambiguous than intended, and we have adjusted the text to be clearer. Our study was mainly based on circulating biomarkers of CD8+ T cell activation. We asked the question whether T cell-mediated inflammation is only a systemic phenomenon or, simultaneously, an active inflammatory process in the heart during AF. Previous biopsy studies have highlighted that the number of CD3+ T cells infiltrating the LA appendage tissue, LA myocardium and surrounding adipose tissue was already higher in patients with paroxysmal AF when compared to patients in sinus rhythm. No data was available about surrogate markers of PAR1- and CD8+ T cell activation in cardiac tissue of patients with AF. It must be mentioned that these studies were performed from tissue obtained by autopsy or cardiac surgery:

  • left atrial appendage (cardiac surgery) Hohmann et al. 2020
  • left atrial appendage (cardiac surgery / autopsy) Wu et al. 2020
  • transmural atrial tissue samples (autopsy) Mitrofanova et al. 2016
  • left atrial appendage (cardiac surgery) Yamashita et al. 2014

Notably, one study found that fibrotic areas of subepicardial fatty infiltrates (of the LA), predominantly contained CD8+ T cells. Post myocardial infarction CD8+ T cells have been shown to mediate adverse remodelling. This, in the opinion of the authors, highlights the ability of CD8+ T cells to migrate to cardiac tissue. Furthermore, AF itself has directs effects on ventricular cardiac function. Given the limited access to human atrial tissue we decided to include data of the right ventricle, which also belongs to the low-pressure system.

In order with your suggestion, we included the specific location of EMB sampling in the figure legend and discussed this important limitation.

Comment 3:

Is the 3rd cohort of patients the same as the first-diagnosed AF patients of cohorts n.1? This should be stated more clearly in the initial description of the population.

Reply:

The reviewer raises an important question. Intervening in the thrombo-inflammatory cascade might be a promising synergistic approach (e.g., anticoagulation + pleiotropic, anti-inflammatory effect on CD8+ T cells) to reduce the disease progression of AF. However, this must be addressed in a prospective trial.

Patients with AF in the third cohort were studied in a retrospective analysis with inherent biases and limitations. We have added this information to the new limitations section and stated more clearly that the third cohort was independent of the first cohort (methods and results). We have also highlighted this directly in the figures.

Comment 4:

More details about those 80 patients [cohort #3] would be appreciated, particularly details on age and specific therapy.

Reply:

We agree. In order with your suggestion, we have added this information to the methods section.

Comment 5:

Each set of results is introduced by the corresponding background (even quoting references) and working hypothesis to lead the reader to understand the process to final conclusion, which is rather unusual. Results are results.

Reply:

We agree. However, we intended to guide a broad readership through the field of thrombo-inflammation. Therefore, we introduced each of the results section with a short background to clarify our main conceptual axis and to explain our selection of analytes.

Comment 6:

The first cohort of patients with newly onset AF is 80 and not 100.

Reply:

It was our intention to state that the whole first cohort consisted of 100 consecutive patients (20 controls and 80 FDAF). To clarify this statement, we have changed the subheading to: ”2.1.1. Patients with first-diagnosed AF and control group”

Reviewer 2 Report

The paper titled: << Cytotoxic CD8+ T cells are involved in the thrombo-inflammatory response during first-diagnosed atrial fibrillation.>> is authored by Julian Friebel et al.

In this research paper, the authors aimed to study the role of Thrombin receptor, protease-activated receptor 1 (PAR1) in the development of the inflammatory profile, including CD8+T lymphocytes, leading to AF. 210 patients were included in this study.

The topic is of interest, but some concerns must be addressed to consolidate the demonstration.

11.      Define FXa-F11a at first use, in the abstract.

22.      Regarding the fact that the authors included an equal number of men and women, in Control and AF groups, they have an excellent opportunity to highlight whether there are differences related to sex in each parameter studied. I suggest to transform the figures to include 4 categories, control male, control female, FDAF male and FAF female.

33.      Table 1 shows that Control and FDAF patients had no differences in CHA2DS2-VASc. May the authors further discuss this fact regarding AF diagnosis and management of these patients?

44.      It is interesting that no significant differences were observed between Control and FDAF patients in terms of Heart failure, cardiac function, Hypertension and Diabetes. This may be a strength in this study to show that CD8+ is directly associated to FDAF. However, some important information must be added in table 1, including: age (yr), body weight (kg), body mass index.

55.      Routinely measured inflammatory biomarkers, such as hs-CRP, BNP, could be added.

66.      Add a limitations section to discuss the complementary experiments that could have been performed, our future perspectives that could be explored to further demonstrate your hypothesis.

77.      Please provide a graphical abstract, a schematic, or proposed mechanism, to summarize the main discovery. It would be interesting to describe a proposed cascade of events involving CD8+T cells, PAR1+T cells, IFN-γ, TNF-a, and FDAF occurrence.

88.      Please provide a highlight section with up to 5-bullet points to describe the main findings of this paper.

Author Response

Reviewer #2:

Comment 1:

Define FXa-FIIa at first use, in the abstract.

Reply:

Thank you for catching this. It has been corrected.

Comment 2:

Regarding the fact that the authors included an equal number of men and women, in Control and AF groups, they have an excellent opportunity to highlight whether there are differences related to sex in each parameter studied. I suggest to transform the figures to include 4 categories, control male, control female, FDAF male and FAF female.

Reply:

The reviewer raises a very interesting question. Gender aspects are very important in the pathogenesis of AM and AF. In order with your suggestion, we now have added a new supplementary figure 1.

Comment 3:

Table 1 shows that Control and FDAF patients had no differences in CHA2DS2-VASc. May the authors further discuss this fact regarding AF diagnosis and management of these patients?

Reply:

Thank you for this suggestion. Saliba et al. has previously shown that the CHA2DS2-VASc score is directly associated with the incidence of new-onset atrial fibrillation and has a relatively high performance for atrial fibrillation prediction. During follow-up (one year), no patient of the control group developed AF or MACE in our study. This, in the opinion of the authors, highlights that CD8+ T cell activation might be one factor that significantly contribute to the onset of AF in a group of vulnerable patients. We have added this information to the description of the control group.

Comment 4:

It is interesting that no significant differences were observed between Control and FDAF patients in terms of Heart failure, cardiac function, Hypertension and Diabetes. This may be a strength in this study to show that CD8+ is directly associated to FDAF. However, some important information must be added in table 1, including: age (yr), body weight (kg), body mass index.

Reply:

We thank the reviewer in supporting our intention. In order with your suggestion, we have added the information to the baseline characteristics (Table 1). Our cohort represents a typical risk constellation for the development of AF.

Comment 5:

Routinely measured inflammatory biomarkers, such as hs-CRP, BNP, could be added.

Reply:

We thank the reviewer for this assistant comment and accordingly added values of BNP, CRP and PCT to the baseline characteristics (Table 1). CRP and PCT plasma levels were below the upper limit of normal. This, in the opinion of the authors, highlights that the new onset of AF in our cohort might not be related to an acute infectious disease.

Comment 6:

Add a limitations section to discuss the complementary experiments that could have been performed, our future perspectives that could be explored to further demonstrate your hypothesis.

Reply:

We totally agree. In accordance with the reviewer’s suggestion, we now have added a limitations section. Although the first diagnosis of AF can be precisely defined, the individual AF duration is obscure. Therefore, based on our data, we cannot conclude at which point of time CD8+ T cell activation is involved in the pathogenesis of AM and the onset of AF. Patients with AF were studied in a retrospective analysis with inherent biases and limitations. Therefore, prospective trials that account for the different phenotypes of AF (or AM as its precursor), as well as the different therapeutic strategies (early rhythm control vs. rate control) and include cardiac magnetic resonance imaging and EMBs would be necessary.

Comment 7:

Please provide a graphical abstract, a schematic, or proposed mechanism, to summarize the main discovery. It would be interesting to describe a proposed cascade of events involving CD8+T cells, PAR1+T cells, IFN-γ, TNF-a, and FDAF occurrence.

Reply:

Thank you! We have revised accordingly.

Comment 8:

Please provide a highlight section with up to 5-bullet points to describe the main findings of this paper.

Reply:

Thank you for this comment. An overview about our main findings has now been added at the beginning of the discussion section.

Round 2

Reviewer 2 Report

The authors provided sufficient improvement of the manuscript.

They replied to all my comments